# T-Cell Immunoglobulin and Mucin Domain 3 (TIM-3) Gene Expression as a Negative Biomarker of B-Cell Acute Lymphoblastic Leukemia

**DOI:** 10.3390/ijms252011148

**Published:** 2024-10-17

**Authors:** Fatemah S. Basingab, Manar Bashanfer, Aisha A. Alrofaidi, Ahmed S. Barefah, Rawan Hammad, Hadil M. Alahdal, Jehan S. Alrahimi, Kawther A. Zaher, Sabah Hassan, Ali H. Algiraigri, Mai M. El-Daly, Saleh A. Alkarim, Alia M. Aldahlawi

**Affiliations:** 1Department of Biological Sciences, Faculty of Science, King Abdulaziz University, Jeddah 21589, Saudi Arabia; 2Immunology Unit, King Fahd Medical Research Center, King Abdulaziz University, Jeddah 21589, Saudi Arabia; 3Hematology Department, Faculty of Medicine, King Abdulaziz University Hospital, King Abdulaziz University, Jeddah 21589, Saudi Arabia; 4Hematology Research Unit, King Fahd Medical Research Center, King Abdulaziz University, Jeddah 21589, Saudi Arabia; 5Department of Biology, Faculty of Science, Princes Nourah bint Abdulrahman University, Riyadh 12211, Saudi Arabia; 6Department of Medical Laboratory Sciences, Faculty of Applied Medical Sciences, King Abdulaziz University, Jeddah 21589, Saudi Arabia; 7Special Infectious Agents Unit-BSL3, King Fahd Medical Research Center, King Abdulaziz University, Jeddah 21589, Saudi Arabia; 8Embryonic Stem Cells Research Unit and Embryonic and Cancer Stem Cells Research Group, King Fahd Medical Research Center, King Abdulaziz University, Jeddah 21589, Saudi Arabia

**Keywords:** acute lymphoblastic leukemia, T-cell immunoglobulin and mucin domain 3, B-cell acute lymphoblastic leukemia, programmed cell death-1, tumor microenvironment, CD4+ T cells, immune checkpoint receptor (ICR)

## Abstract

B-cell acute lymphoblastic leukemia (B-ALL) accounts for 85% of all childhood ALL. Malignancies exhaust T and B cells, resulting in an increased expression of immune checkpoint receptors (ICRs), such as T-cell immunoglobulin and mucin domain 3 (TIM-3). TIM-3 has been found to be dysregulated in different types of cancer. However, there is a lack of rigorous studies on the TIM-3 expression in B-ALL. The current study aimed to measure the expression of TIM-3 at the gene and protein levels and evaluate the potential of TIM-3 as a biomarker in B-ALL. A total of 28 subjects were recruited between 2021 and 2023, comprising 18 subjects diagnosed with B-ALL and 10 non-malignant healthy controls. The B-ALL patients were divided into three groups: newly diagnosed (four patients), in remission (nine patients), and relapse/refractory (five patients). The expression levels of TIM-3 were evaluated using the real-time qPCR and ELISA techniques. The results revealed that the TIM-3 expression was significantly downregulated in the malignant B-ALL patients compared to the non-malignant healthy controls in the mRNA (FC = −1.058 ± 0.3548, *p* = 0.0061) and protein blood serum (*p* = 0.0498) levels. A significant TIM-3 gene reduction was observed in the relapse/refractory cases (FC = −1.355 ± 0.4686, *p* = 0.0327). TIM-3 gene expression allowed for significant differentiation between patients with malignant B-ALL and non-malignant healthy controls, with an area under the curve (AUC) of 0.706. The current study addressed the potential of reduced levels of TIM-3 as a negative biomarker for B-ALL patients.

## 1. Introduction

B-cell acute lymphoblastic leukemia (B-ALL) is the most common type of ALL, accounting for 85% of the total ALL cases [1]. B-ALL is a hematologic malignancy caused by the uncontrolled proliferation of B-lymphocyte precursor cells in the bone marrow [2]. While ALL can affect adults [3], 80% of ALL occurs in children [4]. The annual incidence rate of ALL in the United States is around 3000 cases [5]. In Europe, the overall rate is estimated to be 1.28 per 100,000 people per year, with significant differences linked to age [6]. According to the Saudi Cancer Registry (2020), leukemia is the most common type of childhood cancer in the Kingdom of Saudi Arabia (KSA). Advances in the therapeutic approach have resulted in considerable improvements in overall survival (OS). This improvement is primarily attributed to an aggressive and protracted combination of chemotherapy regimens and, in some cases, hematopoietic stem cell transplantation (HSCT) and, more recently, immunotherapy [7]. Using multiagent chemotherapy has significantly improved the outcomes for pediatric patients, but similar treatments tend to have suboptimal results in adults [8]. With a disease-related mortality rate of around 60%, relapse occurs in around 20%, and the recurrence of B-ALL is the main cause of death [9].

Chronic infections and malignancies frequently exhaust T cells, as seen by the increased expression of immune checkpoint receptors (ICRs) such as cytotoxic T-lymphocyte associated protein 4 (CTLA-4), programmed cell death protein 1 (PD-1), Lymphocyte-activation gene 3 (LAG-3), and T-cell immunoreceptor with immunoglobulin and ITIM domain (TIGIT) [10]. Upon binding to their ligands, these receptors alter T cells to a less efficient profile, reducing proliferation, cytokine production, and cytotoxicity, allowing tumor progression [11]. Targeting some of these ICRs restores the antitumor function and tumor regression, however only in patients with ICR-positive expression [12]. In addition, severe adverse effects have occurred in some of these targeted ICR treatments. Moreover, complete resistance to these ICR-targeted therapies has occurred in various types of tumors [13]. Therefore, the identification of further ICRs that can restore antitumor functions with fewer adverse effects is still required.

Increased T-cell immunoglobulin and mucin domain 3 (TIM-3) expression in human malignancies has been reported. TIM-3 expression was initially observed in interferon (IFN)-producing clusters of differentiation (CD4+) and (CD8+) T cells [14]. TIM-3 is an inhibitory receptor suppressing cytotoxic T cells (CTL) and effector Th1 cell activity [15]. In addition, TIM-3 expression on Th1 cells was shown to be crucial for controlling T-cell tolerance [16]. The expression of TIM-3 was also observed in various cancer cells, endothelial cells (ECs), and tumor-infiltrating lymphocytes (TILs) [17]. TIM-3 has emerged as a promising alternative target for cancer immunotherapy [18]. Inhibiting TIM-3 in combination with other checkpoint receptors in vivo has been demonstrated to boost antitumor immunity and decrease tumor development in several preclinical tumor models [19,20]. In addition, preclinical research indicates that blocking the TIM-3 pathways strengthens T-cell defenses against leukemic cells [20]. Although there is potential, difficulties such as understanding the complicated signaling pathways of TIM-3 and developing specific inhibitors remain unraveled. Recent research has shown that TIM-3 promotes tumor progression through a variety of mechanisms, including facilitating tumor cell migration and invasion, directly suppressing CD4^+^ T cells by activating the IL-6-STAT3 pathway to prevent Th1 polarization, and stimulating the mTOR function in acute myeloid leukemia (AML) cells [21]. However, TIM-3 has an indirect pro-stimulatory effect through the constitutive expression of TIM-3 on antigen-presenting cells (APCs) that can increase costimulatory TIM-3 ligands and cytokines. The study concluded that TIM-3 has opposing roles in innate and adaptive immunity. In innate immunity, TIM-3 promotes inflammation by synergy with Toll-like receptors to induce the production of TNF-α by T helper 1 (Th1), which induces the expression of TIM-3 on activated Th1 that exceeds the expression of TIM-3 on the innate cells. Due to the production of IFN-γ by Th1, the expression of Galectin 9 (Gal9) is induced and interacts with TIM-3, which eventually terminates Th1 [22]. In addition, the performance of natural killer (NK) cells in persons with AML showed that the augmentation of TIM-3 on these cells was related to heightened NK cell activity, eventually leading to improved clinical outcomes in AML patients [23]. New research indicates that AML and other forms of leukemia may respond differently to TIM-3 suppression. However, recent findings suggest that the effects of TIM-3 inhibition may differ between AML and other types of leukemia [24]. Consequently, the potential prognostic significance of the TIM-3 expression patterns in B-ALL remains unknown [11]. TIM-3 evaluation may identify high-risk individuals who require more intense therapy methods more easily, leading to more personalized and effective treatment regimens [17].

A key marker for early therapy and a useful tool for monitoring the development of cancer is the increased expression of ICRs during carcinogenesis [25]. According to the controversial studies on TIM-3 and due to the lack of rigorous studies on the TIM-3 expression in B-ALL, the current study aimed to measure the expression of TIM-3 at the gene and protein levels in human peripheral blood mononuclear cells (PBMCs) from patients with B-ALL categorized as newly diagnosed, in remission, and relapse/refractory compared to healthy controls. In addition, we examined the potential prognostic significance of TIM-3 gene expression and serum TIM-3 in B-ALL.

## 2. Results

### 2.1. Demographic and Clinicopathologic Characteristics of Study Subjects

Table 1 represents the baseline characteristics of the B-ALL patients and non-malignant healthy control subjects. The mean age of the patients was 12.1 ± 14.9 years. Among the B-ALL cohort, 22% of the patients were newly diagnosed, 50% were in remission, and 28% were relapse/refractory cases.

### 2.2. TIM-3 Relative Gene Expression in Malignant B-ALL Subjects Compared to Non-Malignant Healthy Controls

Peripheral blood mononuclear cells (PBMCs) from 18 B-ALL patients and 10 non-malignant healthy controls were collected, and RNAs were purified to evaluate the TIM-3 expression levels. The results presented in Figure 1A show a significant downregulation of the TIM-3 expression among all the B-ALL patients compared to the non-malignant healthy controls (FC = −1.058 ± 0.3548, *p* = 0.0061). The only significant downregulation of TIM-3 expression is reported in the release/refractory patients compared to the non-malignant healthy controls (FC = −1.355 ± 0.4686, *p* = 0.0327) (Figure 1B). No significant differences in the TIM-3 expression were observed in the other two groups, newly diagnosed and in remission, compared to the non-malignant controls. The clinical analyses of the significant TIM-3 downregulation among the relapse/refractory patients are presented in Table 2, in which relapse/refractory patients represent 27.7% of the total B-ALL patients. The clinical results show that the patients’ white blood counts (WBCs) were below 30 units. In addition, 80% of the relapse/refractory patients were at high risk (HR). Moreover, around 60% of the relapse/refractory cases were M1 < 5% blasts and 20% were M2 5–25% blasts of end-of-induction (EOI) bone marrow. Furthermore, 80% of the relapse/refractory patients showed central nervous system 3 (CNS3) status.

### 2.3. Serum TIM-3 Levels in Malignant B-ALL Subjects Compared to Non-Malignant Healthy Controls

Serums from 18 B-ALL patients and 10 non-malignant healthy controls were collected to measure the soluble, circulating TIM-3 levels. A significant reduction in the TIM-3 serum levels was observed in the B-ALL patients compared to the non-malignant healthy controls (*p* = 0.0498), as shown in Figure 2A. No significant differences were reported at the protein levels in the serum TIM-3 levels between the newly diagnosed, remission, and relapse/refractory B-ALL groups and non-malignant healthy controls (Figure 2B). These results indicate that TIM-3 at both the gene and protein levels in the patients with malignant B-ALL was lower than in the non-malignant healthy controls.

### 2.4. Comparisons of TIM-3 at Gene and Protein Levels as Potential Biomarker

Receiver operating characteristic (ROC) curves were built using the TIM-3 gene expression and serum TIM-3 values in patients with malignant B-ALL and non-malignant healthy controls to evaluate the sensitivity and specificity of TIM-3. Although TIM-3 was downregulated in malignant B-ALL compared to the non-malignant healthy controls, the results were used to create ROC curves. The ROC curve analysis revealed that TIM-3 gene expression allowed for significant differentiation between patients with malignant B-ALL and non-malignant healthy controls, with an area under the curve (AUC) of 0.706. In contrast, the AUC of the serum TIM-3 is equal to 0.529 (Figure 3). Therefore, at the gene level alone, TIM-3 may act as a potential negative biomarker in B-ALL.

## 3. Discussion

TIM-3 has emerged as a promising target for cancer immunotherapy, with studies demonstrating that the in vivo blockade of TIM-3 alongside other checkpoint inhibitors improves antitumor immunity and inhibits tumor development in various preclinical tumor models [19]. TIM-3 mRNA and soluble TIM-3 protein have been detected in various tumor tissue samples, and TIM-3 upregulation is linked to the poor prognosis of patients diagnosed with breast cancer, colon cancer, and cervical cancer, among many others [24]. In certain types of leukemia, the gene expression of TIM-3 on malignant cells has been observed. Elevated expression levels of TIM-3 in patients with acute myeloid leukemia (AML) [26] and chronic myeloid leukemia (CML) stem cells have also been observed [27]. Several studies have also detected TIM-3 overexpression on leukemic stem cells (LSCs) but not on healthy stem cells (HSCs) [17]. To the best of our knowledge, TIM-3 gene expression and serum protein levels have not been adequately studied in B-ALL. Published data on TIM-3 activity exist for AML. However, the understanding of the role of TIM-3 in B-ALL is lacking. Given that B-ALL is the most common childhood cancer and the current standard of care for monitoring the disease response uses the minimal residual disease (MRD) by flow cytometry or PCR, which has improved our ability to detect disease relapse before its florid manifestation, there remains an unmet need, especially in relapsed refractory B-ALL, which is to predict which patient cohort will relapse. It is important to be able to understand the predictors of relapse at a biological level, as this will help tailor therapy earlier for those at high risk of relapse. Additionally, current clinical trials are incorporating immunotherapy such as Blinatumomab upfront for standard-risk childhood B-ALL (NCT03914625), which has proven effective. Identifying other potential targets to improve the therapeutic options in childhood B-ALL is vital to be able to reduce the use of conventional chemotherapy, which carries significant short- and long-term side effects. It is unclear whether TIM-3 can be a potential target yet; however, understanding its role is a key first step in such a process. Therefore, in the present study, the gene expression and serum protein levels of TIM-3 were measured from the peripheral blood of B-ALL patients to better understand the role of TIM-3 in the biology of B-ALL.

Our study showed that both the TIM-3 mRNA relative expression levels and its serum protein levels were significantly downregulated in patients with B-ALL compared to non-malignant healthy controls. In contrast to our findings, a study conducted by Balajam et. al. showed that the TIM-3 gene and serum levels were significantly higher in the peripheral blood (PB) and bone marrow (BM) of human patients with acute lymphoblastic leukemia (ALL) compared to healthy controls [28]. Nevertheless, differences in the TIM-3 expression were expected, as these findings were obtained from only newly diagnosed ALL pediatric patients aged 2–10 years, unlike the current study in which PB samples were collected from patients aged 12.1 ± 14.9. In addition, 70% of the patients were diagnosed with B-ALL and 30% were diagnosed with T-lymphoblastic leukemia T-ALL, whereas in our study, patients were diagnosed with B-ALL only [29]. All the patients received similar chemotherapy protocols because the blood samples were collected at remission or relapse before they received any additional treatments. All patients would have received standard-of-care North American leukaemia therapy, which includes multiagent chemotherapy. The standard-risk patients received standard-risk induction with Vincristine, steroids, and peg asparaginase. High-risk patients received high-risk 4 drug induction with Vincristine, steroids, peg asparaginase, and Daunorubicine. The majority of patients in the study had high-risk ALL, only two were standard-risk, and therefore most had received COG HR ALL therapy.

Limited research has been conducted to assess the influence of TIM-3 on B-ALL recurrence, a significant challenge within this leukemia subtype. Liu et al. noted the presence of T-cell exhaustion, marked by the expression of TIM-3 and PD-1, following allogeneic HSCT in cases of B-ALL relapse. However, they were unable to provide a comprehensive explanation of how the increased expression of these immune checkpoints might be linked to the recurrence of the disease. This could occur because of allogeneic stem cell transplantation (HSCT) [30]. In a 2020 study focusing on pediatric B-ALL, Blaeschke and colleagues identified a significant overexpression of TIM-3, individually and in conjunction with PD-1, on CD4+ T cells in the bone marrow. Despite observing no notable differences in the T-cell subpopulations, they recognized TIM-3 as a critical prognostic indicator for an increased risk of relapse in B-ALL patients. This finding emphasizes the potential role of TIM-3 in predicting relapses and underscores the complexity of immune checkpoint interactions in leukemia [24]. However, they were unable to provide a more precise explanation of the relationship between relapse and increased immune checkpoint expression [30].

The downregulation of TIM-3 expression in the current study may be associated with the development or the progression of B-ALL disease. Patients with relapsed/refractory disease showed the significant downregulation of TIM-3 expression compared to non-malignant healthy controls, while those with newly diagnosed or remission status did not show significant differences, which indicates that the lower TIM-3 expression may be linked to more aggressive or treatment-resistant forms of B-ALL disease. These findings also suggest that TIM-3 may play a role, as per our study, in the pathogenesis or progression of relapsed/refractory B-ALL cases. The TIM-3 downregulation in these contexts could contribute to the evasion of immune surveillance by leukemic cells or impair the antitumor immune response. Another explanation of the TIM-3 downregulation in relapsed B-ALL patients is that TIM-3 expression is not static and undergoes dynamic changes in B-ALL. It has been reported that TIM-3 is transiently expressed on activated T cells both in vitro and in vivo due to the various tumor microenvironment cytokines, such as IL-27, a TIM-3 inducer [31].

The immune checkpoints, such as PD-1, PD-L1, CTLA-4, and TIM-3, play an important role in the emerging biomarkers of B-ALL, including disease diagnosis, severity/activity evaluation, or disease outcome prediction. There is increased PD-1 expression on some T-cell subsets in B-ALL patients, especially in relapsed patients [32]. On the other hand, CTLA-4 has been observed in both the chronic and acute types of B-cell lymphocytic leukemia. Further, no evidence was found for chronic T-cell leukemia, which has been reported only in the acute form [33].

The diagnostic worth of the serum levels of pd-1, pd-l1, and CTLA-4 among patients with ovarian cancer was evaluated. The sensitivity and specificity of a PD-1 level of ≥0.568 pg/mL were 67.6% and 90.7%, respectively (*p* = 0.000001), whereas PD-L1 ≥ 0.459 pg/mL showed a sensitivity of 54.1% and a specificity of 93% (*p* = 0.000066). CTLA-4 ≥ 0.595 pg/mL had a 70.3% sensitivity and 90.7% specificity (*p* = 0.000004). When the CTLA4/PD-L1 and PD-1/PD-L1 combinations were used, the sensitivity was very low and equal to 13.5%, whereas the specificity was high at 95.3% [34]. According to our AUC value results, the TIM-3 protein gene is the most promising diagnostic biomarker, in contrast to the serum TIM-3 protein levels. The moderate AUC of the TIM-3 gene expression of 0.706 suggests that probably it could be a relatively efficient negative biomarker that could distinguish patients with B-ALL from healthy individuals. In this context, the low AUC of 0.529 for the serum TIM-3 as an independent negative protein marker for B-ALL is not sufficient.

The current study has several limitations. Obtaining samples that met all three study criteria—newly diagnosed, in remission, and relapse/refractory—proved challenging. A total of 28 samples were collected, consisting of 18 patient samples and 10 healthy controls. The mean age ± SEM of the healthy control subjects was 35 ± 3 years, whereas it was 12.1 ± 14.9 years for the ALL patients. Collecting blood samples from healthy pediatric patients who were not hospitalized was a limitation. Furthermore, it was not possible to rule out the possibility of other underlying medical disorders in pediatric patients admitted to the hospital. Ongoing research at our lab aims to investigate the molecular pathways through which TIM-3 contributes to immune evasion and leukemia progression, focusing on its interaction with other immune checkpoints and cells within the tumor microenvironment. In addition, it is investigating the therapeutic potential of targeting other ICRs, either through direct inhibition or the modulation of their pathways, to enhance the immune system’s ability to recognize and eliminate leukemia cells.

## 4. Materials and Methods

### 4.1. Study Subjects

The study cohort included pediatric and adult patients diagnosed at King Abdulaziz University Hospital between 2021 and 2023. A total of 28 patient samples were included, of which 18 were diagnosed with B-ALL and 10 were non-malignant healthy controls. Patients who had been diagnosed with B-ALL were divided into three groups: (1) newly diagnosed, untreated patients (n = 4), (2) in-remission patients who received therapy as per our institutional standard of care, which is based on the Children’s Oncology Group (CDG) approach involving multiagent chemotherapy, and who had bone marrow evaluations showing complete morphological remission (<5% blasts) (n = 9), or (3) and relapse/refractory patients whose bone marrow or peripheral blood showed circulating blasts following or during first-line therapy (n = 5).

### 4.2. Patients’ Sample Collection

Ethical approval for blood sample collection from the study subjects or their guardians (in the case of pediatric patients) was obtained by the Unit of Biomedical Ethics (UOBE) (Reference No. 512-21) of King Abdulaziz University Hospital (KAUH). All participants or their legal guardians were informed about the study details and asked to sign a consent form. A total of 3–5 mL whole-blood samples were obtained in ethylenediaminetetraacetic acid vacutainer tubes (EDTA BRT; Qiagen, Inc., Manchester, UK) for the TIM-3 mRNA assessment and in Serum Separator Tubes (SSTs) (Becton, Dickinson, NJ, USA) for detecting the serum TIM-3 levels. Peripheral blood mononuclear cells (PBMCs) were isolated and preserved at −80 °C until RNA extraction. The clinicopathological data of the participants were acquired from the Department of Laboratory Medicine and Pathology, KAUH. The baseline characteristics of the B-ALL patients and non-malignant healthy control subjects are presented in Table 1.

### 4.3. TIM-3 Gene Expression

Relative gene expression of TIM-3 was analyzed using quantitative reverse transcription polymerase chain reaction (RT-qPCR). This technique is conducted by performing the following procedures: RNA extraction and nucleic acid concentration and purity measurement, complementary DNA (cDNA) synthesis, primer design, and RT-qPCR.

#### 4.3.1. RNA Extraction and Nucleic Acid Concentration and Purity Measurement

Total RNA was extracted from the isolated PBMCs using the RNeasy Mini Kit ™ blood RNA kit (Qiagen, Hilden, Germany) following the manufacturer’s instructions. Steps involved using various kit reagents and RNA-free water, following the outlined protocol at 4 °C. The concentration and the purity of the extracted RNA were measured using a NanoDrop 2000c spectrophotometer (Thermo Fisher Scientific, Waltham, MA, USA) at a 260/280 ratio~2.

#### 4.3.2. Complementary DNA (cDNA) Synthesis

The process began by standardizing the amount of RNA by adding nucleic acid-free water. Next, the RNA samples were denatured using the thermocycler set at 70 °C for 5 min. Subsequently, a master mix was prepared utilizing the GoScript™ Reverse Transcriptase kit (Promega, Madison, WI, USA), followed by transferring it to the thermal cycler apparatus for 140 min. The produced cDNA was stored at −20 °C for downstream qPCR application.

#### 4.3.3. Real-Time Quantitative PCR (RT-qPCR)

RT-qPCR was used to evaluate the expression levels of two selected genes: TIM-3 and the housekeeping gene glyceraldehyde-3-phosphate dehydrogenase (GAPDH). For PCR amplification of cDNA samples, the primer sequences of the TIM-3 and endogenous glyceraldehyde-3-phosphate dehydrogenase (GAPDH) genes were designed using the NCBI Genome Browser to acquire the sequences of the genes of interest. The sequences of primers were as follows: TIM-3 F: ggagatgtgtccctgaccat; TIM-3 R: aaggctgcagtgaagtctct; GAPDH F: ccagaacatcatccctgcct; GAPDH R: cctgcttcaccaccttcttg.

The RT-qPCR reactions for TIM-3 and GAPDH using all samples were processed in duplicate in 96-well plates using Bio-Rad iQ SYBR Green Supermix and a CFX Connect™ Real-Time PCR device (Bio Rad Laboratories, Inc., Hercules, CA, USA) according to the manufacturer’s instructions. The RT-qPCR program was conducted using a single initial cycle of 10 min at 95 °C, followed by 40 amplification cycles of 15 s at 95 °C, and 1 min at 60 °C. The amplified products were verified at the end of each cycle and their purity was determined by analyzing the melting curves. Ct values were used to calculate the mean Ct. The expression levels of TIM-3 were normalized to the housekeeping gene and calculated using the 2^−ΔΔ^ Ct method [35].

### 4.4. TIM-3 Serum Levels

A commercially available Human TIM-3 Enzyme-Linked Immunosorbent Assay (ELISA) Kit (Solarbio^®^, Beijing, China) (cat. no. SU-bn10232) was used to determine the serum levels of TIM-3. According to the manufacturer’s instructions, 50 μL of either standard or room-temperature serum samples were added to a microtiter plate, followed by the addition of 100 μL of enzyme conjugate. The plates were incubated for 60 min at 37 °C. Plates were then washed 4 times before the addition of 50 μL of both substrates A and B and were incubated for 15 min at 37 °C. Finally, 50 μL Stop Solution was added, followed by measuring the optical density (OD) at 450 nm using a microtiter plate reader.

### 4.5. Statistical Analysis

GraphPad Prism 10.0.3 (GraphPad Software, Inc., La Jolla, CA, USA) was used for statistical analyses of the relative gene expression of TIM-3 and the serum TIM-3 levels, where *p* ≤ 0.05 was considered statistically significant. Non-parametric tests were chosen if the included variables were categorical or had a skewed distribution accordingly. Significant changes in the gene expression between the non-malignant controls and ALL samples were noted using an unpaired, two-tailed *t*-test. Additionally, one-way ANOVA (two-tailed Mann–Whitney and Kruskal–Wallis tests) was applied for selected parameters to compare the three groups. Results are presented as mean ± standard error of the mean (SEM).

## 5. Conclusions

This study investigated the TIM-3 expression at the mRNA level and its protein serum levels in patients diagnosed with B-ALL. In our study, the TIM-3 expression was significantly reduced in the patients with malignant B-ALL compared to the non-malignant healthy controls, especially in the relapsed/refractory cases. Lower levels of TIM-3 expression may be linked to the onset or progression of B-ALL, particularly in more aggressive diseases. This study examined the potential of TIM-3 as a negative differential biomarker for B-ALL, suggesting that significantly lower levels of TIM-3 gene expression could be used as a negative biomarker, especially in the relapsed/refractory phase.

## Figures and Tables

**Figure 1 ijms-25-11148-f001:**
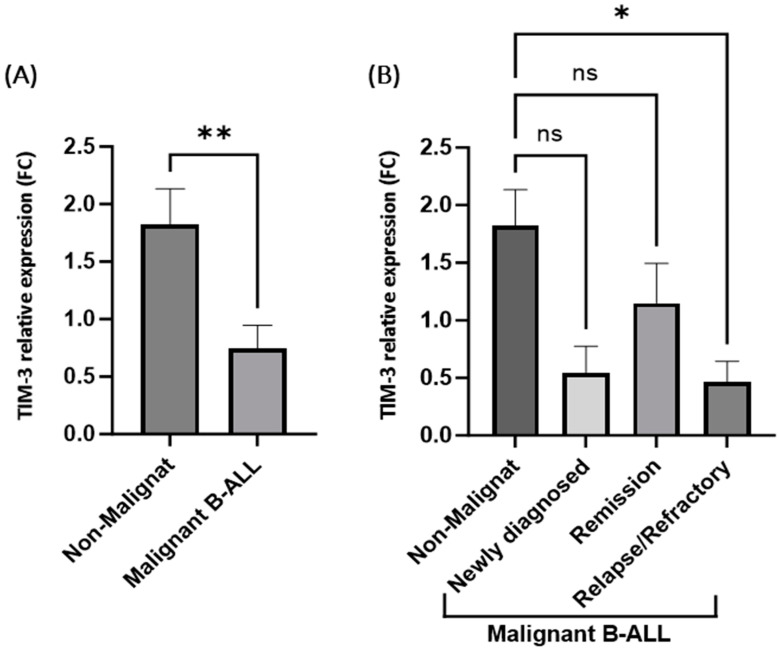
TIM-3 relative gene expression in patients with B-ALL compared to non-malignant healthy controls. (**A**) Purified RNAs were isolated from peripheral blood mononuclear cells (PBMCs), RT-qPCR determined the gene expression of TIM-3, and the expression of GAPDH was normalized. Shows the expression of TIM-3 in all study subjects: 10 non-malignant healthy controls and 18 patients with malignant B-ALL. ** Represents a significant difference between groups with *p* = 0.0061 (using unpaired *t*-test). (**B**) Shows the expression of TIM-3 in non-malignant healthy controls and patients with malignant B-ALL that were categorized into newly diagnosed, remission, relapse, and refractory. * Represents a significant difference between groups with *p* = 0.0327 (using one-way ANOVA, multiple comparisons), and ns shows the non-significant differences between groups. Statistical analyses were conducted using GraphPad Prism 10.0.3.

**Figure 2 ijms-25-11148-f002:**
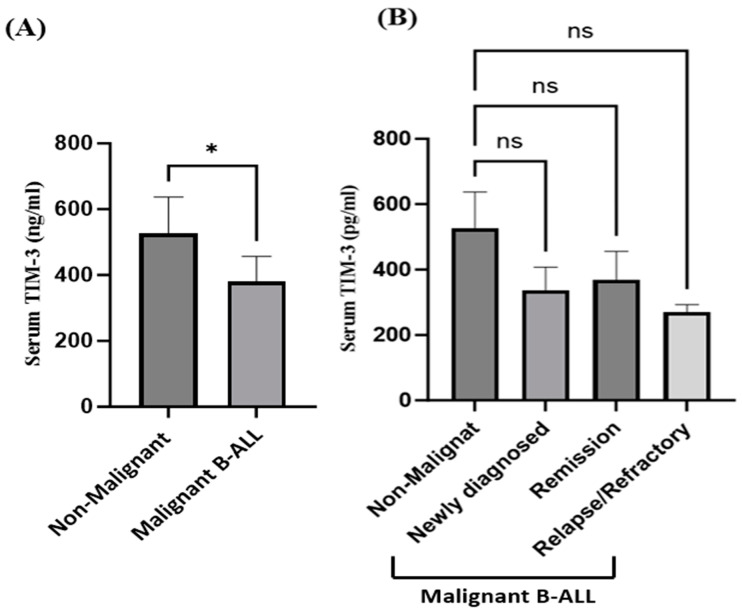
The levels of serum TIM-3 in patients with malignant B-ALL compared to non-malignant healthy controls. (**A**) Serums from the peripheral blood of all study subjects were used to detect the TIM-3 serum levels by ELISA Kit. * Shows significant differences between groups with *p* = 0.0498 (using Mann–Whitney *t*-test). (**B**) Serums from newly diagnosed B-ALL, the in-remission phase, and relapse/refractory were used to measure TIM-3 serum levels by ELISA Kit. Ns: no significant differences between groups. Statistical analyses were conducted using GraphPad Prism 10.0.3.

**Figure 3 ijms-25-11148-f003:**
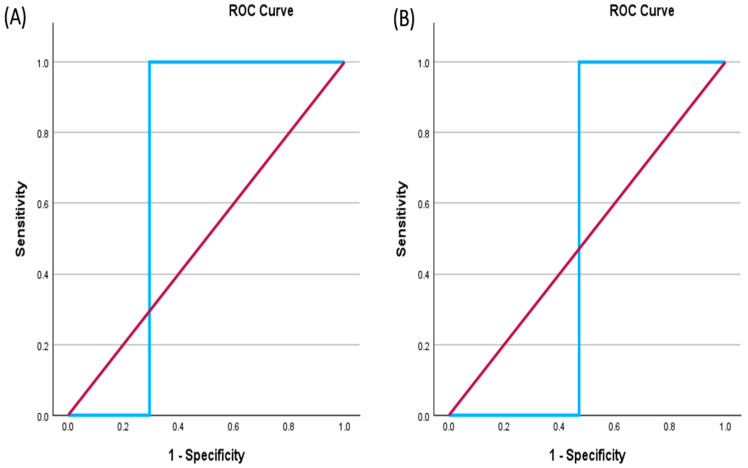
Receiver operating characteristic curves (ROC) for TIM-3 gene expression and serum TIM-3. (**A**) Represents the TIM-3 gene expression where the AUC = 0.706. This indicates that the TIM-3 gene expression has a reasonable diagnostic ability, with a moderate balance between sensitivity and specificity. (**B**) Shows the serum TIM-3 levels where the AUC = 0.529; AUC: area under the curve. This suggests that the serum TIM-3 levels may not be a reliable biomarker for diagnosis. The red lines of the ROC represent no discrimination (AUC = 0.5), whereas the blue lines show the AUC of the results.

**Table 1 ijms-25-11148-t001:** Baseline characteristics of malignant B-cell acute lymphocytic leukemia (B-ALL) and non-malignant healthy control subjects.

	Total				Non-Malignant				Malignant B-ALL			
Parameters	Mean ± SEM		Median	IQR	Mean ± SEM		Median	IQR	Mean ± SEM	Median		IQR
**Number of participants, n (%)**	28 (100)				10 (36)				18 (64)			
**Gender**	-				6 Female (60%), 4 Male (40%)				10 Female (56%), 8 Male (44%)			
**Age (years)**	21.6 ± 17.2		15	29	35 ± 3		38	12.5	12.1 ± 14.9	8		7.8
**Initial WBC count (K/uL)**	-		-	-	-		-	-	91.6 ± 207.7	8.4		71.6
**Disease status at study enrollment**												
**Newly diagnosed**				**In Remission**					**Relapse/Refractory**			
**4 (22%)**				**9 (50%)**					**5 (28%)**			
**Male**		**Female**		**Male**		**Female**			**Male**		**Female**	
**2**		**2**		**4**		**5**			**1**		**4**	

**Table 2 ijms-25-11148-t002:** The clinical characteristics of relapse/refractory B-ALL patients at initial diagnosis that showed significant differences when compared to non-malignant healthy controls.

Relapse/Refractory (Number of Relapse/Refractory = 5, Representing 27.7% of Total B-ALL Patients)				
**Initial WBC count at diagnosis (K/uL)**	**≤5**	**5–30**	**≥30**	
	40% (WBC: 2.4–4.5)	40% (WBC: 8.8–13)	20% (WBC: 226)	
**COG risk stratification at initial diagnosis**	**SR**	**HR**	**Not available**	
	0	80%	20%	
**End-of-induction (EOI) bone marrow response**	**M1 < 5% blasts**	**M2 5%–25% blasts**	**M3 > 25% blasts**	**Not available/not performed**
	60%	20%	20%	0
**Central nervous system (CNS) status at initial diagnosis**	**CNS1**	**CNS2**	**CNS3**	
	20%	0	80%	

## Data Availability

The original contributions presented in the study are included in the article, further inquiries can be directed to the corresponding author.

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
