# Peer review of "T-Cell Immunoglobulin and Mucin Domain 3 (TIM-3) Gene Expression as a Negative Biomarker of B-Cell Acute Lymphoblastic Leukemia"

_ijms, 2024, doi:10.3390/ijms252011148_

Round 1

Reviewer 1 Report

Comments and Suggestions for Authors

I have the following concerns:

1. a. B-ALL patients were divided into three groups: newly diagnosed (4 patients), in remission 30 (9 patients), and relapse/refractory (5 patients). Were the various chemotherapy regimens used influence the statistical analysis in R/R patients  and in remission patients? Are there chemotherapy regimens used statistically affecting the levels of expression of TIM-3 and if so, what are these regimens?

b. Please mention analytically all the chemotherapy regimens used. 

2. The methodology is ok, but tiring for the reader. Please condense it to include the more important elements.

3. What is the clinical significance of your study? Are there relevant clinical trials evaluating drugs targeting TIM-3 for B-ALL?

4. What in your opinion is the biggest restriction, the biggest disadvantage of your study? Please state that to the discussion.  

Author Response

First of all, all authors would like to express their gratitude to all reviewers for their time and effort in reviewing our manuscript to be in the best form. We sincerely thank the reviewers for their valuable comments, which we will do our best to address point by point. All changes are highlighted in yellow. Thank you again for your help and support.

  1. a. B-ALL patients were divided into three groups: newly diagnosed (4 patients), in remission 30 (9 patients), and relapse/refractory (5 patients). Were the various chemotherapy regimens used influence the statistical analysis in R/R patients and in remission patients? Are there chemotherapy regimens used statistically affecting the levels of expression of TIM-3 and if so, what are these regimens? b. Please mention analytically all the chemotherapy regimens used.

Answer: All the patients received similar chemotherapy protocols because the blood samples were collected at remission or relapse before they received any additional treatments.  All patients would have received standard-of-care North American leukaemia therapy which includes multi agent chemotherapy. The standard-risk patients received standard-risk induction with Vincristine, steroids, and peg asparaginase. High-risk patients received high-risk 4 drug induction with Vincristine, steroids, peg asparaginase, and Daunorubicine. The majority of patients in the study were High risk ALL, only 2 were a standard risk, therefore most have received the COG HR ALL therapy. (Page 7, lines 234- 241)

It is important to clarify there are no variations in chemotherapy given to leukemic patients that may affect the result of TIM-3 expression because the blood samples were collected at diagnosis, at remission, or relapse.

  1. The methodology is ok, but tiring for the reader. Please condense it to include the more important elements.

Answer: Agreed, methodology is now condensed as required. 

  1. What is the clinical significance of your study? Are there relevant clinical trials evaluating drugs targeting TIM-3 for B-ALL?

Answer: Agreed, Published data on TIM-3 activity exist for AML. However, the understanding of the role of TIM-3 in B-ALL is lacking. Given that B-ALL is the most common childhood cancer and the current standard of care for monitoring disease response uses minimal residual disease (MRD) by flow cytometry or PCR, which has improved our ability to detect disease relapse before its florid manifestation. There remains an unmet need, especially in relapsed refractory B-ALL, which is to predict which patient cohort will relapse. It is important to be able to understand predictors of relapse at a biological level, as this will help tailor therapy earlier for those at high risk of relapse. Additionally, current clinical trials are incorporating immunotherapy such as Blinatumomab upfront for standard-risk childhood B-ALL (NCT03914625), which has proven effective. Identifying other potential targets to improve therapeutic options in childhood further B-ALL is vital to be able to reduce the use of conventional chemotherapy, which carries significant short- and long-term side effects. It is unclear if TIM-3 can be a potential target yet, however, understanding its role is a key first step in such a process. (Page 8, lines 203- 216)

  1. What in your opinion is the biggest restriction, the biggest disadvantage of your study? Please state that to the discussion.

Answer: Agreed, restriction and the biggest disadvantage are stated in the discussion. (Page 9, lines 281-299)

Reviewer 2 Report

Comments and Suggestions for Authors

The manuscript entitled: “T-cell Immunoglobulin and Mucin Domain 3 (TIM-3) Gene Expression as A  Negative Biomarker of in B-cell Acute Lymphoblastic Leukemia(ID: ijms-3244415) by Basingab et al. aims to analyze the expression of TIM-3 in PBMsc in newly diagnosed, remission and in relapsed/refractory B-ALL patients.

Albeit the review is well written and of interest, comments should be addressed to further improve the manuscript.

Comments:

1.    Result section: within the ROC analysis an AUC of 0.706 is rather moderate and should not be too enthusiastic. Moreover, please provide the cut-off of TIM-3 determined by the ROC analysis. In addition, please provide a description of the ROC analysis also within the statistical section.

2.    Discussion section: the discussion section should be balanced according to strength and weaknesses of the analysis. Moreover, what is the rationale for the analysis of TIM-3 in B-ALL pts. Please clarify and highlight the potential benefit for this pts cohort (e.g. what are the exact recommendations for the clinicians). Discussion section: line 194-196: this sentence should be revised, since the results of this analysis provide not a better understanding of the biology of B-ALL overall.

Author Response

First of all, all authors would like to express their gratitude to all reviewers for their time and effort in reviewing our manuscript to be in the best form. We sincerely thank the reviewers for their valuable comments, which we will do our best to address point by point. All changes are highlighted in yellow. Thank you again for your help and support.

  1. Result section: within the ROC analysis an AUC of 0.706 is rather moderate and should not be too enthusiastic. Moreover, please provide the cut-off of TIM-3 determined by the ROC analysis. In addition, please provide a description of the ROC analysis also within the statistical section.

Answer: Agreed, the data were re-analyzed, and the same results were obtained. The AUC of 0.706 is moderate but is considered acceptable in many publications. For example, according to Manderkar (2010) who stated that “In general, an AUC of 0.5 suggests no discrimination (i.e., ability to diagnose patients with and without the disease or condition based on the test), 0.7 to 0.8 is considered acceptable, 0.8 to 0.9 is considered excellent, and more than 0.9 is considered outstanding(1). In addition, similar results
was generated and puplished from our lab (2).

  1. Mandrekar, J. N. (2010). Receiver operating characteristic curve in diagnostic test assessment. Journal of Thoracic Oncology5(9), 1315-1316.
  2. Aldahlawi, A., Basingab, F., Alrahimi, J., Zaher, K., Pushparaj, P. N., Hassan, M. A., & Al-Sakkaf, K. (2023). Herpesvirus entry mediator as a potential biomarker in breast cancer compared with conventional cytotoxic T‑lymphocyte‑associated antigen 4. Biomedical Reports19(2), 56.

The smallest cutoff value is the minimum observed test value minus 1, and the largest cutoff value is the maximum observed test value plus 1. All the other cutoff values are the averages of two consecutive ordered observed test values.

  1. Discussion section: the discussion section should be balanced according to strength and weaknesses of the analysis. Moreover, what is the rationale for the analysis of TIM-3 in B-ALL pts? Please clarify and highlight the potential benefit for this pts cohort (e.g. what are the exact recommendations for the clinicians). Discussion section: line 194-196: this sentence should be revised, since the results of this analysis provide not a better understanding of the biology of B-ALL overall.

Answer: Rationale for using TIM-3 in B-ALL. Published data on TIM-3 activity exist for AML. However, the understanding of the role of TIM-3 in B-ALL is lacking. Given that B-ALL is the most common childhood cancer and the current standard of care for monitoring disease response uses minimal residual disease (MRD) by flow cytometry or PCR, which has improved our ability to detect disease relapse before its florid manifestation. There remains an unmet need, especially in relapsed refractory B-ALL, which is to predict which patient cohort will relapse. It is important to be able to understand predictors of relapse at a biological level, as this will help tailor therapy earlier for those at high risk of relapse. Additionally, current clinical trials are incorporating immunotherapy such as Blinatumomab upfront for standard-risk childhood B-ALL (NCT03914625), which has proven effective. Identifying other potential targets to improve therapeutic options in childhood further B-ALL is vital to be able to reduce the use of conventional chemotherapy, which carries significant short- and long-term side effects. It is unclear if TIM-3 can be a potential target yet, however, understanding its role is a key first step in such a process. (Page 8, line 203- 219)

Reviewer 3 Report

Comments and Suggestions for Authors

Basingab and collaborator investigated the potential role of TIM-3 as a negative biomarker for B-ALL patients. The study is well designed and enrolled a total of 28 patient samples (18 affected by B-ALL and 10 control).

Overall the study is well structured, the introduction and results are clear as well discussions and conclusions. 

In order to accept the study I just suggest to make few correction:

1-Figures are out of focus. Please use tiff file and check the size (in particoular fig 1 and Fig 2 seems to be different).

2-I suggest to remodulate the sentence in duscussion line 208-212. The use of CAR-T to target TIM 3 is challenging mainly due to the TIM-3  high expression in normal tissue, so I suggest to not enphasize its targeting with immunotherapy. 

3- Using of SYBR Green is a quite old fashion techniques. Are you able to provide similar results with TaqMan Assay or clarify why did you prefear to use SYBR instead TaqMan assay?

Comments on the Quality of English Language

Quality of English is fine. 

Author Response

First of all, all authors would like to express their gratitude to all reviewers for their time and effort in reviewing our manuscript to be in the best form. We sincerely thank the reviewers for their valuable comments, which we will do our best to address point by point. All changes are highlighted in yellow. Thank you again for your help and support.

Comments and Suggestions for Authors

Basingab and collaborator investigated the potential role of TIM-3 as a negative biomarker for B-ALL patients. The study is well designed and enrolled a total of 28 patient samples (18 affected by B-ALL and 10 control). Overall the study is well structured, the introduction and results are clear as well discussions and conclusions. In order to accept the study I just suggest to make few correction:

1-Figures are out of focus. Please use tiff file and check the size (in particoular fig 1 and Fig 2 seems to be different).

Answer: Done

2-I suggest to remodulate the sentence in duscussion line 208-212. The use of CAR-T to target TIM 3 is challenging mainly due to the TIM-3  high expression in normal tissue, so I suggest to not enphasize its targeting with immunotherapy.

Answer: Agreed with this, TIM3 cannot be used as a CAR-T target.

3- Using of SYBR Green is a quite old fashion techniques. Are you able to provide similar results with TaqMan Assay or clarify why did you prefear to use SYBR instead TaqMan assay?

Answer: Agreed that TaqMan assay is more advanced than SYBR green in which with the three known quantitation assays: one-step RT-PCR RNA, Two-step RT-PCR RNA, and DNA quantitation, it can go further in allelic discrimination. As the focus of this paper is RT-PCR RNA, SYBR Green I dye can detect all double-strand DNA accurately. In addition, this method is well-established and well-optimized in our lab. We will consider the use of TaqMan in our next paper. 

Round 2

Reviewer 1 Report

Comments and Suggestions for Authors

I have no further concerns. 

Reviewer 2 Report

Comments and Suggestions for Authors

The manuscript entitled: “T-cell Immunoglobulin and Mucin Domain 3 (TIM-3) Gene Expression as A  Negative Biomarker of in B-cell Acute Lymphoblastic Leukemia(ID: ijms-3244415) by Basingab et al. aims to analyze the expression of TIM-3 in PBMsc in newly diagnosed, remission and in relapsed/refractory B-ALL patients.

After revision of the manuscript, the authors addressed all my initial comments sufficiently.